# The Gaze Patterns of Group Fitness Instructors Based on Different Levels of Training and Professional Experience

**DOI:** 10.3390/sports11080153

**Published:** 2023-08-15

**Authors:** Francisco Campos, Catarina M. Amaro, João P. Duarte, Rui Mendes, Fernando Martins

**Affiliations:** 1Instituto Politécnico de Coimbra, Escola Superior de Educação de Coimbra, 3030-329 Coimbra, Portugal; rmendes@esec.pt (R.M.); fmlmartins@esec.pt (F.M.); 2Instituto de Telecomunicações, Delegação da Covilhã, 6201-001 Covilhã, Portugal; 3Research Unit for Sport and Physical Activity (CIDAF), Faculty of Sport Sciences and Physical Education, University of Coimbra, 3040-256 Coimbra, Portugal; catarinammamaro@gmail.com (C.M.A.); joaopedromarquesduarte@gmail.com (J.P.D.)

**Keywords:** gaze pattern, fixation, saccade, group fitness instructor, training, experience

## Abstract

The way in which group fitness instructors observe participants has a great influence on their pedagogical intervention. Based on the above, the main objective of this research is to characterize and compare their gaze patterns according to their training and professional experience. Twenty group fitness instructors of choreographed classes participated, aged between 18 and 42 years old, and for the comparison, four groups were created. Eye movements were captured with TOBII Pro Glasses 3, and data were coded and analyzed using the TOBII Pro Lab software. For the characterization of the gaze patterns, descriptive statistics were used in terms of count (f/m) and duration (s/m), while the comparison was performed using a one-way ANOVA test. More trained and experienced instructors tend to look at participants less, in count (80.59 ± 0.74) and duration (17.74 ± 0.71), with significant differences between the groups in some areas of interest (head, lower body, and other). There are also significant differences in the total number of eye fixations (*F* = 34.614; *p* = 0.001; *η*^2^ = 0.866; effect size very high). In conclusion, and projecting future works, it is important to understand how these gaze patterns are related to pedagogical behaviors in general or based on some specific factors (e.g., pedagogical feedback).

## 1. Introduction

The eyes are one of the sensory organs of the brain that work in a complementary way to produce vision from anything in the visual field (e.g., objects and persons) [1]. Humans are mostly vision-oriented, and it is estimated that 90% of all processed sensory stimuli are visual [2]. The use of vision to capture information from the visual field allows for people to adapt to the surrounding environment and respond according to the main characteristics of each situation. It is common to read and hear that the human eye is the organ of vision. Perhaps, it is more correct to say that it is in the eye that vision begins its journey through the human visual system [3,4]. According to these authors, we look with the eyes, but we see with the brain.

The eye-tracking system (ETS) allows for the study of eye position and optical movement [5,6]. With ETS, it is possible to assess where an individual looks, how long and how many times, and variables that are affected by cognitive processes beyond attention (e.g., perception and decision making). It is accepted that eyes reflect the mental processes of what we are looking at any given moment [7]. The majority of research using ETS focuses on fixations and saccades [8,9]. “Whereas fixations enable focus on a particular object, saccadic movements facilitate shifts of attention from one location to another” [10] (p. 250). Land [11] affirms that fixations are the gaps between saccades in which the gaze is almost stationary, and it is when the processed visual stimulus has a greater resolution. On the other hand, during one saccade, there is no information processing. For Land and Hayhoe [12], about 90% of the viewing time is spent on fixations.

The fixation time is not consensual in the literature (e.g., 80–150 ms [1], 200–300 ms [5], and 100–600 ms [10]). In a recent study on sports areas (basketball), the authors assume that the “fixation has to be at least 100 ms long” [13] (p. 2). Some of the most common fixation metrics used are the number (total number and specific number according to different areas of interest (AOI)) and the duration (also total and per AOI [14]). In many eye-tracking investigations, “researchers want to know how long or how often participants looked at a particular part of a stimulus” [7] (p. 53). When this is one of the main goals under study, different AOI should be created to understand specifically where the observation is directed.

In sports, individual performance and collective performance inherently depend on the efficient visual tracking of the relevant information associated with the realized task [15]. Looking at the right place and at the right time is particularly important regardless of the sport (e.g., football [16], cricket [17], field hockey [18], volleyball [19], judo [20], climbing [21], and basketball [13]). The main areas of research that use eye movement monitoring technology in the sports field are the following [8,22]:(a)An analysis of the role of the visual system in motor skills formation and development (e.g., successful or unsuccessful trials, vision, and attention prior to skill execution).(b)A comparison of eye movement patterns, among athletes with different mastering levels, in technical and tactical aspects (e.g., visual strategies and spatial awareness).

In fitness and gym activities, motor skills and performance, among other objectives (e.g., self-esteem and weight loss), in a general perspective, are equally relevant for participants and instructors. Group activities were one of the most sought-after services by gym goers before the pandemic that occurred in early 2020 [23]. Naturally, due to the need to maintain public health, these activities were canceled and, later, adapted to this new reality. Gradually, they re-entered user preferences and market trends projected for the sector, both internationally [23,24] and specifically in Portugal [25]. This is one factor, among others, that should concern and be an object of reflection of the market’s main actors, including the fitness instructor themself. The instructor has a preponderant role due to their impact on the participant’s perception [26]. The way they provide instruction [27] or pedagogical feedback [28], for example, can make a difference in how participants recognize what is, for them, a good instructor, that is, an instructor with quality [26].

The way the instructor observes the participants or the class as a whole can have a great influence on their pedagogical intervention, which is making decision making (e.g., correcting the move or waiting for a new execution before the intervention) based on what they see, think, and decide to be the best for a particular situation. Training and experience also have an influence on performance in the competitive sports field [1,16,29,30] and especially in fitness group activities [31,32]. Based on the above, the main objective of this investigation is to characterize and compare group fitness instructors’ gaze patterns according to their training and professional experience.

Recently, there has been an amplification of the research focus. Not only athletes, but also other sports agents, such as referees and coaches, have been studied [22]. For example, Damas e Ferreira [33] investigated the visual fixation patterns of two groups of basketball coaches with different skills. If there are differences in this context, it is possible that in noncompetitive sport activities, group fitness activities (e.g., aerobics), the same can be observed. Perhaps group fitness instructors, according to their training and professional experience, look at their participants in practice in a different way when compared with novices and maybe respond more effectively, providing instruction and/or pedagogical feedback at the right moment with the right purpose, for example. The use of this high technology (ETS) provides innovative data, which can be useful in the search for answers for a market in expansion. With these data, fitness instructors will be better aware of how they observe their class and, with this, be able to reflect on their own behavior, adapting and improving it. Training professionals should also consider these data in order to provide better observation skills to fitness instructors. That will allow them a better pedagogical intervention, the improvement of the provided service and, consequently, the participant’s satisfaction.

## 2. Materials and Methods

### 2.1. Participants

Twenty group fitness instructors of a choreographed class (aerobics), aged between 18 and 42 years old ($\bar{x}$ ± SD = 23.95 ± 6.49), 10 female (23.10 ± 5.87) and 10 male (24.80 ± 7.26), participated in this study. Being an exploratory investigation, a convenience sample was used, respecting the criteria presented below for comparison. Hence, according to the training and professional experience, four groups were created:Group 1 (G1): Instructors in Training (IiT), attending their 2nd year of a bachelor’s degree program, with 39 h of specific training in aerobics (*n* = 6; 3 f, 19.83 ± 1.16 years old).Group 2 (G2): IiT, attending their 3rd year of the same degree program, with a further 26 h of step and 26 h of water aerobics (*n* = 6; 3 f, 21.67 ± 3.14 years old).Group 3 (G3): Instructors with Training (IwT), newly graduated from the same bachelor’s degree program, without professional experience, only with 240 h of intervention in internships during a school year, in addition to the specific training previously specified (*n* = 4; 2 f, 22.00 ± 0.81 years old).Group 4 (G4): Senior IwT (*n* = 4; 2 f, 35.50 ± 4.50 years old), with experience as an instructor of fitness group activities (13.75 ± 4.50 years).

### 2.2. Instruments

The eye movements were captured with TOBII Pro Glasses 3 eye tracking glasses (ETG) (Stockholm, Sweden), and all data were coded and analyzed with TOBII Pro Lab software (version 1.217).

### 2.3. Procedures

After the definition of the profile of each group under analysis based on training and professional experience (G1, G2, G3, and G4), the instructors were invited to participate, being informed about the task, protocol, and aim of the investigation. Some information was given to standardize the session: warm-up phase; approximate duration of three to six minutes; music with 130 to 140 beats per minute (bpm); total pyramid or pairs pyramid choreographic methodology [34]; spatial organization (instructor and participants) previously defined and marked on the ground (Figure 1); and same level (beginner) class (20 participants).

Immediately before data collection, both the instructors and the participants were reminded of the procedures. After that, individual calibration of the ETG was made before the start of the proposed task.

### 2.4. Analysis

For the characterization of the gaze patterns, descriptive statistics ($\bar{x}$ ± SD) was used to analyze count [fixations per minute (f/m)] and duration [seconds per minute of class (s/m)]. The results were presented in a total (T) perspective, regardless of the location of the fixation, and stratified via AOI. For that, two main AOI were defined: participant (P) and other (O) (e.g., empty space between two participants, window, sound system). The participants’ AOI are further subdivided into three categories: head (H), upper body (UB), and lower body (LB). It assumed fixation time of at least 100 ms, as in Marques et al. [13].

The comparison between groups was performed using a one-way ANOVA test after validation of its assumptions of normality and homogeneity [35]. For samples below 30, the assumption of normality was verified using the Shapiro–Wilk test [35]. When normality was not verified, the analysis of symmetry was resorted, using the following condition: skewness/standard error skewness ≤ 1.96 [36].

Levene’s test was used to verify the assumption of homogeneity. To carry out multiple comparisons, the Tukey HSD post hoc test was used if the assumptions of normality and homogeneity were verified. When the assumption of homogeneity was not verified, the post hoc Games–Howell test was used [35]. The classification of the effect size (*η*^2^) in the one-way ANOVA test was carried according to Ferguson [37]: very high (*η*^2^ > 0.50); high (0.25 < *η*^2^ ≤ 0.50); medium (0.05 < *η*^2^ ≤ 0.25); and small (*η*^2^ ≤ 0.05). Statistical data analysis was performed using IBM SPSS 28 for a significance level of 5% (*p* < 0.05).

## 3. Results

Table 1 presents the results of the characterization of the gaze patterns for fixations, considering count (f/m) and duration (s/m), as well as the comparison between the groups. The average duration of the sessions under analysis is a few more than four minutes (241.70 s).

In terms of count, more trained and experienced instructors present lower values of total f/m (G1: 104.83 ± 6.71; G2: 100.92 ± 2.48; G3: 89.68 ± 1.70; G4: 80.59 ± 0.74), with significant differences (*F* = 34.614; *p* = 0.001; *η*^2^ = 0.866; effect size very high) between G4 and G1; G4 and G2; G4 and G3; G3 and G1; and G3 and G2. The same tendency (lower values in more trained and experienced instructors) happens in LB (G1: 20.83 ± 1.12; G2: 18.91 ± 1.43; G3: 16.18 ± 3.77; G4: 9.32 ± 1.35), with significant differences (*F* = 28.639; *p* = 0.001; *η*^2^ = 0.843; effect size very high) between G4 and G1; G4 and G2; G4 and G3; and G3 and G1. The same tendency is observed in O (G1: 39.50 ± 8.58; G2: 34.92 ± 7.87; G3: 21.17 ± 3.01; G4: 10.42 ± 3.45), with significant differences (*F* = 17.950; *p* = 0.001; *η*^2^ = 0.771; effect size very high) between G4 and G1; G4 and G2; G3 and G1; and G3 and G2. In the AOI H, the opposite happens: more trained and experienced professionals present higher values (G1: 31.44 ± 2.98; G2: 32.93 ± 4.55; G3: 35.60 ± 1.07; G4: 41.36 ± 3.26), with significant differences (*F* = 7.367; *p* = 0.002; *η*^2^ = 0.589; effect size very high) between G4 and G1 and G4 and G2. In the P (*F* = 0.818; *p* = 0.503; *η*^2^ = 0.133; effect size medium) and UB (*F* = 2.575; *p* = 0.090; *η*^2^ = 0.326; effect size high), there are no significant differences between the analyzed groups.

From the analysis of the fixations’ duration, it is possible to verify significant differences in the AOI H, LB, and O. In H, more trained and experienced instructors present superior values (G1: 6.89 ± 1.30; G2: 7.18 ± 1.65; G3: 7.82 ± 0.85; G4: 11.34 ± 1.03), with significant differences (*F* = 10.843; *p* = 0.001; *η*^2^ = 0.670; effect size very high) between G4 and G1; G4 and G2; and G4 and G3. In both LB (G1: 5.15 ± 1.20; G2: 4.91 ± 0.78; G3: 4.19 ± 0.36; G4: 2.31 ± 0.31) and O (G1: 5.16 ± 0.54; G2: 4.95 ± 1.16; G3: 3.22 ± 1.04; G4: 2.04 ± 0.51), there is a decrease in the values with the increase in training and professional experience. Differences occurs in G4 and G1; G4 and G2; and G4 and G3 (LB: *F* = 10.907; *p* = 0.001; *η*^2^ = 0.672; effect size very high) and G4 and G1; G4 and G2; G3 and G1; and G3 and G2 (O: *F* = 13.477; *p* = 0.001; *η*^2^ = 0.716; effect size very high). In T (*F* = 1.565; *p* = 0.237; *η*^2^ = 0.227; effect size medium), P (*F* = 0.061; *p* = 0.980; *η*^2^ = 0.011; effect size small), and UB (*F* = 1.872; *p* = 0.175; *η*^2^ = 0.260; effect size high), there is no significant differences.

Briefly, differences mainly occur between G4 and G1 [count (f/m): H, LB, O, T; duration (s/m): H, LB, O] and G4 and G2 [count (f/m): H, LB, O, T; duration (s/m): H, LB, O]. There are also significant differences in G4 and G3 [count (f/m): LB, T; duration (s/m): H, LB], G3 and G1 [count (f/m): LB, O, T; duration (s/m): O] and G3 and G2 [count (f/m): O, T; duration (s/m): O].

In P and UB, unlike the other categories, count and duration do not follow the same trend. In both P (f/m: G1: 65.39 ± 3.48; G2: 65.99 ± 7.49; G3: 68.50 ± 4.67; G4: 70.16 ± 4.13; s/m: G1: 16.49 ± 3.27; G2: 16.29 ± 4.08; G3: 16.09 ± 0.45; G4: 15.70 ± 0.94) and UB (f/m: G1: 13.11 ± 1.07; G2: 14.14 ± 6.09; G3: 16.72 ± 1.13; G4: 19.48 ± 3.72; s/m: G1: 4.44 ± 1.97; G2: 4.20 ± 2.18; G3: 4.07 ± 0.41; G4: 2.05 ± 0.71), there is an increase in number of f/m in more trained and experienced instructors, but the time of the fixations is lower with more training and professional experience.

Complementarily, it is presented the heat maps, in terms of count and duration, of the fixations and saccades, gaze patterns (Figure 2). With this, it is possible to analyze the data also in terms of *where*.

In both cases, count and duration a high similarity between the G1 and G2 are verified, with the differences becoming more accentuated between groups from G3 onwards. Between G1 and G2, the main visible difference occurs in the core zone, which is the zone with greater incidence with a higher coverage in G2. From G3 onwards, it is possible to verify a greater scope of action, mainly in the count, with the instructors presenting a larger coverage area regarding the observation of the class. However, while in G3, this observation continues to be very directed towards the center of the group, specifically for LB, in G4, this observation becomes more open, covering most of the participants, even those who are in the extremities of the group.

To understand better the difference between instructors with high levels of training and professional experience, the heat maps [count (f/m)] of the gaze patterns of two instructors (one from G1 and another from G4) are presented as an example in Figure 3.

## 4. Discussion

As stated by Panchuk et al. [1] (p. 178), “eye movements differ between experts and novices across a range of skills”. Campbell and Moran [29] found significant differences between elite and novice golfers in the green analysis before the putt, suggesting that professional golfers used more economical gaze patterns (fewer fixations but longer duration). In this present study, the tendency between count and duration is the same; more fixations during more time or less fixations during equally less time.

More trained and professionally experienced group fitness instructors present more fixations in participant’s H and present less in participant’s LB, O, and T. The same trend happens when the focus analysis is the duration of fixations. Although aerobics is a group fitness activity, it is a class made up of different participants who like and want to be treated according to their specificities; thus, instructors should adapt their sessions accordingly [26]. Eye-to-eye communication, looking at the participant when providing pedagogical feedback [28], whether for correction or motivation, will certainly be more effective than if you are looking into the void, not facing the participants when information is transmitted. Training and professional experience could interfere with this particular variable. In fact, more trained and experienced instructors adapt and adopt different behaviors [32], and those skills, competencies, and years of practice allow them to act differently, providing a class more centered on the individual characteristics of each participant. If the instructor looks more at the participant’s head/face, naturally, will look less for another participant’s AOI, such as LB.

More trained and professional experienced instructors also look less for AOIs O and T. As previously referred, if the focus of the instructors is on the participants, naturally, they will be able to keep it on them more easily, looking fewer times at less relevant aspects of the class (e.g., empty space between participants, windows, sound system), classified as O in this present study. As for the fact that in total, a smaller number of fixations with shorter durations occur in more trained and experienced instructors, which can happen because they do not need to stare at something or at each participant in specific situations to interpret and define strategies to solve any problem that may arise. They can, for example, present a higher number of saccades making use of their peripheral vision [38] to interpret the situation and act in line with the need.

Although not presenting significant differences, the results of P and, particularly UP, deserve our attention. More trained and experienced instructors look more often at P and UB but for less time. The opposite occurs with less trained and experienced ones (look less often at the P and UB but for a longer time). This can happen because, with experience, it is not necessary to take so much time to analyze and interpret the situation, acting with more effectiveness [29].

The heat maps of the gaze patterns presented in Figure 3 are extreme examples. However, in a way, it demonstrates how an instructor observes the class at the beginning of their career and how, over some years, this observation begins to take place. These differences were also highlighted in previous studies [1,16,29,30], although not in sport technical intervention, but related to the athlete’s sport performance. In this particular case, a larger observation allows for a better conscience of what happens during the class, making the intervention more appropriate and more effective.

The use of high technology (such as ETS) to showcase these innovative findings in the unexplored domain of fitness (specifically the eye movements of fitness instructors) is the main strength of this research. With these data, it is possible to better understand how the group fitness instructors observe their class and, with that, relate the observation with, for instance, their pedagogical behaviors [31] or their pedagogical feedback [32]. However, we must be careful when interpreting and extrapolating the results to other contexts. The reduced number of participants, and the analysis focused on the warm-up phase in a specific activity (aerobics), are the main limitations of this study, which should be considered in future research.

## 5. Conclusions

In conclusion, eye movements differ between experts and novices group fitness instructors, as stated by Panchuk et al. [1], in other domains. More trained and experienced instructors present less fixations in count and duration, but this trend is different in some AOI. Also, the fixations localization is different. More trained and experienced instructors present a broader view, allowing for a more efficient analysis of the entire class.

The obtained results raise some questions that will be the object of study in future works. Why do more trained and experienced group fitness instructors present fewer fixations compared with novice professionals? With training and experience, there is no need to fixate on participants to make decisions using more saccadic movements and, with that, the peripheral vision, as suggested in Tenenbaum and Bar-Eli [38]. Is it possible that, during a saccade, there may be information processing?

In the future, it will be also important to understand the relationship between instructors’ gaze patterns and their efficiency in pedagogical intervention. This present study allows us to understand how the group fitness instructors look and where they look. However, it will be equally or more relevant to understand how these gaze patterns are related to some pedagogical aspects of their intervention during the session in general [39] or in specific variables (e.g., pedagogical feedback [32], modeling [40]).

## Figures and Tables

**Figure 1 sports-11-00153-f001:**
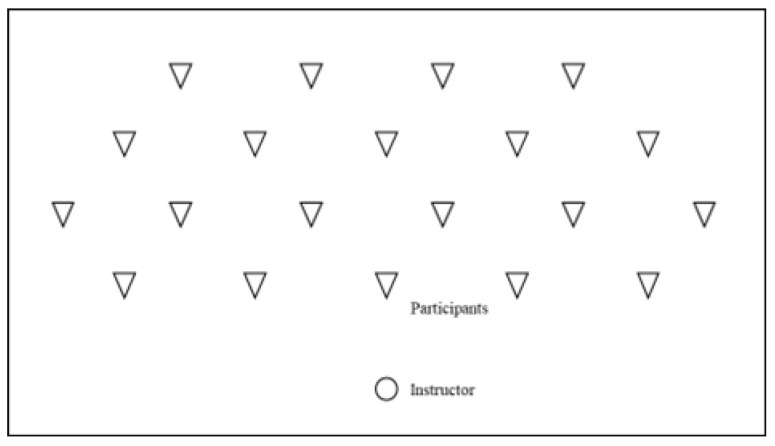
Spatial organization of the class (instructor (circle) and participants(triangles)).

**Figure 2 sports-11-00153-f002:**
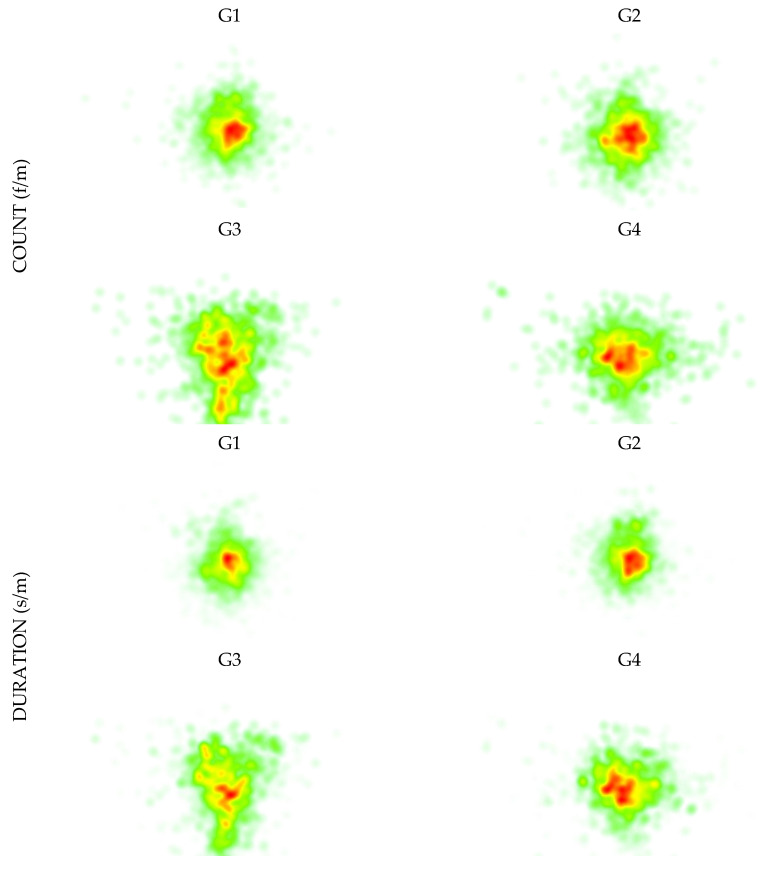
Heat maps of the fixations and saccades in count and duration.

**Figure 3 sports-11-00153-f003:**
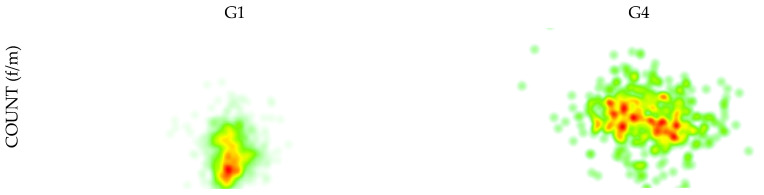
Heat maps of the gaze patterns of an instructor from G1 and another from G4.

**Table 1 sports-11-00153-t001:** Characterization and comparison of the gaze patterns (fixations).

	AOI	G1	G2	G3	G4	*F*	*p*	$\eta_{}^{2}\;$
COUNT (f/m)	P	65.39 ± 3.48	65.99 ± 7.49	68.50 ± 4.67	70.16 ± 4.13	0.818	0.503	0.133
	H	31.44 ± 2.98 ^a^	32.93 ± 4.55 ^b^	35.60 ± 1.07	41.36 ± 3.26 ^a,b^	7.637	0.002 *	0.589
	UB	13.11 ± 1.07	14.14 ± 6.09	16.72 ± 1.13	19.48 ± 3.72	2.575	0.090	0.326
	LB	20.83 ± 1.12 ^a,d^	18.91 ± 1.43 ^b^	16.18 ± 3.77 ^c,d^	9.32 ± 1.35 ^a,b,c^	28.639	0.001 *	0.843
	O	39.50 ± 8.58 ^a,d^	34.92 ± 7.87 ^b,e^	21.17 ± 3.01 ^d,e^	10.42 ± 3.45 ^a,b^	17.950	0.001 *	0.771
	T	104.83 ± 6.71 ^a,d^	100.92 ± 2.48 ^b,e^	89.68 ± 1.70 ^c,d,e^	80.59 ± 0.74 ^a,b,c^	34.614	0.001 *	0.866
DURATION (s/m)	P	16.49 ± 3.27	16.29 ± 4.08	16.09 ± 0.45	15.70 ± 0.94	0.061	0.980	0.011
	H	6.89 ± 1.30 ^a^	7.18 ± 1.65 ^b^	7.82 ± 0.85 ^c^	11.34 ± 1.03 ^a,b,c^	10.843	0.001 *	0.670
	UB	4.44 ± 1.97	4.20 ± 2.18	4.07 ± 0.41	2.05 ± 0.71	1.872	0.175	0.260
	LB	5.15 ± 1.20 ^a^	4.91 ± 0.78 ^b^	4.19 ± 0.36 ^c^	2.31 ± 0.31 ^a,b,c^	10.907	0.001 *	0.672
	O	5.16 ± 0.54 ^a,d^	4.95 ± 1.16 ^b,e^	3.22 ± 1.04 ^d,e^	2.04 ± 0.51 ^a,b^	13.477	0.001 *	0.716
	T	21.65 ± 3.42	21.25 ± 4.40	19.31 ± 0.59	17.74 ± 0.71	1.565	0.237	0.227

* significant for *p* < 0.05; multiple comparison with post hoc test: (a) G4 vs. G1. (b) G4 vs. G2. (c) G4 vs. G3. (d) G3 vs. G1. (e) G3 vs. G2; f/m = fixations per minute; s/m = seconds per minute of class; AOI = area of interest; P = participant; H = head; UB = upper body; LB = lower body; O = other; T = total.

## Data Availability

All data can be obtained by contacting the corresponding author.

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
