# Peer review of "The Gaze Patterns of Group Fitness Instructors Based on Different Levels of Training and Professional Experience"

_sports, 2023, doi:10.3390/sports11080153_

Round 1
Reviewer 1 Report
Comments to the Author:
Regarding the evaluation of the manuscript entitled "The gaze patterns of the Aerobics fitness instructors in different 2 levels of training and professional experience"
· Please describe how was the sample size calculated?
· Page 2, Lines 52-53: Please clarify the point you are attempting to make. "When this is one of the main goals under study, different AOI 52 should be created, considering the investigation`s specificity."
· The idea behind this work is good but discussion needs precision and clarity. Could you elaborate more on this by specifically talking about trained and professional experienced Aerobics fitness instructors present more 188 fixations in participants H and present less in participants LB, O and T. The same trend 189 happens when the focus analysis is the duration of fixations.
Moderate editing of English language required
Reviewer 2 Report
General comments
The authors have clearly stated that the purpose of the study was to characterize and compare the gaze patterns of group fitness instructors, according to their training and professional experience. The paper is well-written, easy to follow and adds merit to the vital role of the way group fitness instructors provide instruction or pedagogical feedback. Given this approach, this work can enhance future attempts in similar research area. However, I have highlighted a few suggestions and concerns in my specific comments section (below) that need to be addressed before accepting this work for publication.
Specific comments
TITLE & THROUGHOUT THE MANUSCRIPT
- Change “aerobics” to “group” fitness instructors throughout the manuscript. This particular area of fitness services has been extensively evolved and therefore, the term “aerobics” does bot perfectly reflect the broad spectrum of group fitness classes in the global health and fitness industry.
ABSTRACT
- Lines 13-14: Change “20” to “Twenty” and state the type of group fitness classes you investigated (i.e., choreographed).
- Lines 17-18: Remove the sentence “for a significance level of 5% (p < .05).
- Exact percentage of change (Δ) and effect sizes (η2) should be added in results.
INTRODUCTION
- Please add the relevant paper to strengthen the statement about the popularity of group exercise at the national, regional, and global level.
Suggested references:
Batrakoulis A, Veiga OL, Franco S, et al. Health and fitness trends in Southern Europe for 2023: A cross-sectional survey. AIMS Public Health. 2023; 10(2): 378-408.
RESULTS
- Exact percentage of change should be presented in results either in text or tables.
DISCUSSION
- Strengths and limitations (e.g., small sample size) should be stated in the same paragraph at the end of the discussion section.
Moderate editing is needed.
Reviewer 3 Report
Thank you for the opportunity to review this article.
The article is interesting, but needs some additions and improvements.
Recommendations:
Introduction - I recommend highlighting the novel aspects of this study and its impact on the teaching methods of fitness trainers.
Section 4 of Discussions should be added after section 3. Results to highlight the impact that the study results have on previous studies.
The conclusions require their revision to be more concise and clear in relation to the relevance of the results.
Round 2
Reviewer 3 Report
The authors improved the manuscript according with my recommendations.